# Anticancer Potential and Molecular Targets of Pristimerin in Human Malignancies

**DOI:** 10.3390/ph17050578

**Published:** 2024-04-30

**Authors:** Kirti S. Prabhu, Serah Jessy, Shilpa Kuttikrishnan, Farina Mujeeb, Zahwa Mariyam, Ummu Habeeba, Nuha Ahmad, Ajaz A. Bhat, Shahab Uddin

**Affiliations:** 1Translational Research Institute, Academic Health System, Hamad Medical Corporation, Doha 3050, Qatar; serahjessym@gmail.com (S.J.); skuttikrishnan@hamad.qa (S.K.); v-zpurayil@hamad.qa (Z.M.); v-uhabeeba@hamad.qa (U.H.); nuha001ahmad@gmail.com (N.A.); 2Department of Biosciences, Integral University, Lucknow 226026, Uttar Pradesh, India; farinam@iul.ac.in; 3Department of Human Genetics-Precision Medicine in Diabetes, Obesity, and Cancer Program, Sidra Medicine, Doha 26999, Qatar; abhat@sidra.org; 4Dermatology Institute, Academic Health System, Hamad Medical Corporation, Doha 3050, Qatar; 5Laboratory of Animal Research Center, Qatar University, Doha 2713, Qatar

**Keywords:** Pristimerin, apoptosis, autophagy, reactive oxygen species, signaling pathways

## Abstract

The growing global burden of malignant tumors with increasing incidence and mortality rates underscores the urgent need for more effective and less toxic therapeutic options. Herbal compounds are being increasingly studied for their potential to meet these needs due to their reduced side effects and significant efficacy. Pristimerin (PS), a triterpenoid from the quinone formamide class derived from the Celastraceae and Hippocrateaceae families, has emerged as a potent anticancer agent. It exhibits broad-spectrum anti-tumor activity across various cancers such as breast, pancreatic, prostate, glioblastoma, colorectal, cervical, and lung cancers. PS modulates several key cellular processes, including apoptosis, autophagy, cell migration and invasion, angiogenesis, and resistance to chemotherapy, targeting crucial signaling pathways such as those involving NF-κB, p53, and STAT3, among others. The main objective of this review is to provide a comprehensive synthesis of the current literature on PS, emphasizing its mechanisms of action and molecular targets with the utmost clarity. It discusses the comparative advantages of PS over current cancer therapies and explores the implications for future research and clinical applications. By delineating the specific pathways and targets affected by PS, this review seeks to offer valuable insights and directions for future research in this field. The information gathered in this review could pave the way for the successful development of PS into a clinically applicable anticancer therapy.

## 1. Introduction

Cancer is a significant global health issue that has attracted attention due to its devastating impact on humans worldwide. Chemotherapy, radiotherapy, immunotherapy, genetic and targeted treatment, hormone therapy, and surgery represent subsets of the multifaceted approaches that have been employed in the pursuit of cancer treatment. Among the myriad therapeutic alternatives at our disposal, chemotherapy stands out as a widely recognized modality that exhibits remarkable efficacy in the management of neoplastic growths [1]. Nevertheless, it is crucial to recognize the possible drawbacks linked to chemotherapy, including the development of multidrug resistance (MDR) in cancer cells and the occurrence of severe drug-induced adverse effects. Henceforth, it is imperative to foster the development of more efficacious treatment modalities [2,3].

For centuries, traditional Chinese medicine (TCM) has been seen as a potential remedy for the challenges that cancer poses [4]. This choice is primarily driven by the remarkably diminished likelihood of encountering adverse effects and an elevated probability of achieving triumphant outcomes. Numerous naturally occurring compounds discovered within the realm of TCM have been scientifically validated for their remarkable anticancer attributes. Among these compounds are resveratrol, dioscin, berberine, curcumin, baicalein, wogonin, silibinin, quercetin, celastrol, and PS [1,4,5,6,7,8,9].

Triterpenoids are a class of essential compounds found in plants as specialized metabolites. Quinone methide triterpenoids, a specific group of compounds containing a quinone ring, have been of particular interest due to their cytotoxic and anti-tumoral activities [10]. Pristimerin (PS), 20α-3-hydroxy-2-oxo-24-nor-friedela-1-10,3,5,7-tetraen-carboxylicacid-29-methylester, with the molecular formula C_30_H_40_O_4_ (Figure 1), a quinonemethide triterpenoid compound of indigenous origin, is taxonomically categorized within the families Celastraceae and Hippocrateaceae. In 1951, Bhatnagar and Divekar achieved the commendable feat of isolating PS from botanical specimens of *Pristimerae indica* and *P. grahami*. Consequently, in 1954, diligent efforts were made by Kulkarni and Shah to successfully ascertain its precise molecular structure, thus solidifying its scientific identity [11]. Pristimerin has various pharmacological properties including anticancer, antioxidant, anti-inflammatory, antibacterial, antimalarial, and insecticidal actions [11]. Recent pharmacokinetic prediction studies employing molecular docking and simulation methodologies indicate that PS exhibited a substantial binding energy of −10.9 kcal/mol when interacting with pri-miR-378a [12]. The molecular dynamics simulation provided evidence of a stable connection between the PS and miRNA complex [12]. PS has the potential to regulate cancer-causing miRNAs, making it a promising candidate for cancer prevention and treatment due to its precise control of miRNA activity. PS’s therapeutic value has drawn the interest of an increasing number of researchers who have made important contributions to the field. These characteristics have been extensively reported in various scientific studies [1,11].

PS demonstrates remarkable pharmacological efficacy in combating tumors through its capacity to selectively target and modulate a wide array of signal transduction pathways and molecular targets. These pathways encompass a myriad of intricate molecular interactions, such as the phosphatidylinositol 3-kinase (PI3K)/AKT pathway, the nuclear factor-kappaB (NF-kB) pathway, the reactive oxygen species (ROS)/mitogen-activated protein kinase (MAPK) pathway, the sonic hedgehog (Shh)/glioma-associated oncogene homolog 1 (Gli1) pathway, and the hypoxia-inducible factor 1a (HIF-1a)/sphingosine kinase 1 (SPHK1) pathway. Moreover, it significantly impacts the complex interaction between proteasome functioning and telomerase activity. PS exerts its anticancer effects via the induction of G1 phase arrest, apoptosis initiation, and autophagy facilitation in cancer cells. Furthermore, it induces inhibitory effects on the tumor cells through the modulation of cellular adhesion, reduction in cancer stem cell (CSC) activity, inhibition of the epithelial–mesenchymal transition (EMT), and inhibition of angiogenesis [11,13].

Several investigations have demonstrated that treating cancer cells with PS can reverse chemoresistance. Multiple studies have also shown that PS can increase the therapeutic effectiveness of various chemotherapeutic agents [14,15,16]. The primary aim of this review is to systematically consolidate the existing knowledge on the anticancer potential of PS, detailing how it interacts with and influences various molecular pathways to inhibit cancer progression. This review aims to provide a comprehensive resource for both researchers and clinicians, offering a clearer understanding of the multifunctional roles of PS in cancer therapy. By highlighting well-established and newly discovered mechanisms of PS, this synthesis encourages further research into its clinical applications and the development of novel therapeutic strategies that may incorporate or enhance the efficacy of PS.

This paper introduces several new scientific aspects of PS research. It examines the less explored molecular interactions of PS, particularly its ability to modulate proteasome functionality and telomerase activity. These areas are seldom the primary focus of cancer therapy studies but have shown promise in preliminary investigations. Additionally, this review discusses how PS has the potential to overcome chemoresistance, which is a major hurdle in current cancer treatment patterns. By elucidating the molecular basis for PS’s ability to reverse resistance mechanisms, this work contributes to the frontier of cancer research. It suggests innovative approaches for integrating PS into existing treatment regimens, which not only deepens our understanding of PS’s therapeutic scope but also establishes a foundational framework for future translational and clinical research in oncology.

## 2. Induction of Apoptosis

Apoptosis is a genetically regulated form of programmed cell death essential for eliminating unnecessary, damaged, or infected cells, thus maintaining cellular homeostasis [17]. The two main categories of apoptosis are the extrinsic death receptor system and the intrinsic mitochondrial pathway [18].

The death receptor pathway involves the activation of death receptors, such as Fas and tumor necrosis factor (TNF) receptor-1. This process results in the formation of a death-inducing signaling complex composed of the adaptor protein, the Fas-associated death domain, and initiator caspases, specifically caspase-8. In the context of the mitochondrial pathway, increased mitochondrial permeability results in the release of cytochrome-c, Apaf-1, and other apoptotic components into the cytoplasm. This event is followed by caspase activation and subsequent cellular death [1,11].

Several reports have documented that PS induces cell death both in vitro and in vivo (Table 1 and Table 2) via the activation of mitochondrial signaling, caspase activation (both extrinsic and intrinsic), and PARP cleavage [5,19,20,21,22]. In addition, PS has been shown to induce apoptosis via the production of reactive oxygen species (ROS), resulting in significant cytotoxic effects across a diverse range of cancers [5,13,23,24,25,26].

## 3. Induction of Autophagy

Autophagy is a critical process involved in the destruction of impaired organelles and aged proteins, as well as in the preservation of cellular homeostasis [50]. Autophagy in cancer biology serves a dual function, encompassing both tumor promotion and inhibition. The induction of autophagy in response to diverse cellular stressors plays a crucial role in regulating cell death, therefore offering a promising avenue for developing novel anticancer therapeutics [11,50]. ATG proteins encoded by autophagy-related genes (ATGs) help build the autophagosome [51]. The LC3 conversion (from LC3-I to LC3-II), which is recognized as a hallmark of autophagy [52], occurs when ATG4B combines with ATG7 to produce LC3II, which is formed by the conjugation of LC3-I and phosphatidylethanolamine. In cancer related to esophageal and cholangiocarcinoma, PS enhanced the ratio of LC3-II/LC3-I and increased the accumulation of autophagosomes [53]. Multiple in vitro investigations have demonstrated an upregulation of Beclin-1, ATG7, LC3-II, and p62 expression levels in malignancies associated with breast cancer (BC) and chronic myeloid leukemia (CML) [28,33,54]. Furthermore, the concurrent administration of PS and paclitaxel enhanced extracellular signal-related kinase (ERK)-mediated autophagic cell death. This was evidenced by an increase in the degradation of p62 and the expression of beclin1 [15]. In contrast, the study conducted by Zhang et al. [47] showed that PS had inhibitory effects on autophagy in lung cancer A549 and NCI-H446 cells. This was achieved by downregulating LC3BII and beclin1, ultimately enhancing the apoptotic response induced by cisplatin.

## 4. Inhibition of Cell Migration, Invasion and Metastasis

The metastasis of cancer encompasses a range of intricate cellular and molecular events, including cancer cell invasion, migration, and the establishment of metastatic colonies in clinical settings [11,55].

### 4.1. Epithelial–Mesenchymal Transition (EMT)

The inhibition of the epithelial–mesenchymal transition (EMT) involves the transformation of epithelial cells into mesenchymal cells, thereby enhancing the migratory and invasive capabilities of tumor cells [56,57]. Previous studies have demonstrated that treatment with PS significantly reduces the expression of key EMT markers including N-cadherin, fibronectin, vimentin, ZEB1, and Snail in both prostate and lung cancer cells, confirming its role in modulating EMT pathways [58,59]. Furthermore, the research by Shu et al. revealed that PS inhibits EMT in trophoblast cells via the miR-542-5p/EGFR signaling axis [60]. This effect of PS extends to suppressing proliferation, migration, and the EMT in trophoblast cells, while simultaneously inducing apoptosis. Notably, the downregulation of miR-542-5p, AGO2, and EGFR expression in these cells contributes to this phenomenon. Moreover, the co-treatment of PS with miR-542-5p silencing exhibited a synergistic effect, highlighting the potential of PS as a therapeutic agent for preventing embryo implantation in cases of ectopic pregnancy. In studies involving MDA-MB-231 triple-negative breast cancer cells, PS was shown to inhibit cell proliferation and revert EMT by upregulating E-cadherin and downregulating N-cadherin [61]. Additionally, PS suppresses the expression of integrin β3 mRNA and protein, which are crucial components of the heterodimeric transmembrane receptor associated with the EMT. Experimental data indicate that silencing integrin β3 enhances the anti-EMT effects of PS, whereas overexpressing this receptor diminishes its efficacy. In xenograft models, PS treatment led to the notable suppression of tumor growth, underscoring its potential utility in cancer therapy.

### 4.2. Cell Adhesion and Cytoskeleton Inhibition

Matrix metalloproteinases (MMPs) play a pivotal role in carcinoma development by modulating the tumor microenvironment. Notably, the proteins MMP2 and MMP9, which are crucial for facilitating invasion and metastasis, are downregulated by PS in a dose-dependent manner. This effect was specifically observed in EC9706 and EC109 cell lines, which are models for esophageal cancer, indicating the targeted action of PS on these enzymes [62,63]. Furthermore, PS has demonstrated significant efficacy in reducing lung cancer cell viability, migration, invasion, and capillary formation. These effects are mediated through the downregulation of the EphB4/CDC42/N-WASP signaling pathway. This pathway is critical for orchestrating mitochondrial-mediated intrinsic apoptosis and endoplasmic reticulum (ER) stress, thereby impeding cancer cell survival and spread [48]. Additional research involving the NCI-H1299 lung cancer cell line has shown that PS not only suppresses cell proliferation but also induces apoptosis. Alongside these effects, there was a notable reduction in the migration and invasion capabilities of H1299 cells, further supporting the potential of PS as a multifaceted therapeutic agent in the treatment of lung cancer [58].

## 5. Cancer Stem Cells (CSCs)

The inhibition of cancer stem cells (CSCs) is pivotal in understanding and managing carcinogenesis, tumor metastasis, and resistance to chemotherapy. Specific CSC markers such as CD44, aldehyde dehydrogenase (ALDH), and CD133 are integral to the processes of carcinogenesis, self-renewal, and therapy resistance. Additionally, transcription factors like NANOG, SOX-2, and OCT-4 have been implicated in the processes of metastasis and invasion, highlighting their roles in tumor progression and cellular invasiveness [64,65]. However, a significant challenge in CSC research is the identification of a precise and specific panel of CSC markers, as current markers often lack the specificity required for effective targeting and analysis. The natural compound PS has shown promising results in reducing the proportion of ALDH+ cells and inhibiting tumorsphere formation, suggesting its potential in targeting cellular mechanisms essential for CSC maintenance. In the context of uveal melanoma (UM), treatment with PS was found to decrease the levels of stemness-associated proteins such as Slug and Sox2, although the levels of Nanog and KLF4 remained unchanged, indicating selective effects on stem cell-associated pathways [64]. In prostate cancer, using the PC-3 cell line as a model, PS administration resulted in a reduction in the levels of CD44, CD133, and other stemness factors including KLF4, OCT4, and AGO2. This indicates its broad-spectrum activity against various markers associated with stemness [64]. Additionally, the efficacy of PS in eradicating CSCs in esophageal squamous cell carcinoma (ESCC) has been linked to its modulation of the NF-κB signaling pathway, further underscoring its therapeutic potential [64]. PS also demonstrates a potent anticancer effect against breast cancer stem cells, inducing cell death through both apoptosis and incomplete autophagy. The underlying mechanisms involve the inhibition of thioredoxin-1, activation of the ASK1 and JNK signaling pathways, and generation of reactive oxygen species (ROS), all of which are crucial to the PS-induced cell death process [28,54]. These findings support the role of PS as a multifaceted agent capable of targeting and modifying the behavior of cancer stem cells across different cancer types.

## 6. Angiogenesis Inhibition

Angiogenesis, the process of new blood vessel formation, is critical for various physiological functions but can contribute detrimentally to cancer development when aberrantly activated. In parallel, the formation of new lymphatic vessels, known as lymphangiogenesis, also plays a significant role in cancer progression [66]. Consequently, therapies that target these processes, particularly anti-angiogenic treatments, are crucial in the fight against cancer. Recent research has highlighted the efficacy of PS in inhibiting angiogenesis triggered by vascular endothelial growth factor (VEGF) in human umbilical vascular endothelial cells (HUVECs). This inhibition has been observed both in vitro and in vivo, marking PS as a potent inhibitor of angiogenic signaling pathways [67]. Further studies have shown that in hypoxic conditions, PS effectively suppresses the expression and phosphorylation of key regulatory proteins such as HIF-1α, SPHK-1, and AKT/GSK-3β in PC-3 prostate cancer cells. This suppression leads to a consequential decrease in VEGF levels and angiogenesis, underlining the potential of PS in targeted cancer therapies [68]. Additionally, PS has demonstrated significant anticancer activity in uveal melanoma cells by inhibiting TNFα-induced p65 translocation, IκBα phosphorylation, and NF-κB-dependent gene expression. These molecular interventions contribute to the reduced cell viability and diminished clonogenic, migration, and invasion capabilities of uveal melanoma cells. Remarkably, when PS is combined with vinblastine, a primary therapeutic agent, a synergistic effect is observed, enhancing the therapeutic outcome. Moreover, PS induces apoptosis in these cells, providing a multi-faceted approach to combat uveal melanoma [69].

## 7. PS and Targeted Pathways

Over the past several decades, extensive research has revealed that many chronic health issues stem from the dysregulation of genes primarily involved in cell cycle control, particularly in oncology. This dysregulation often leads to unchecked cellular proliferation and subsequent metastasis. PS, a naturally occurring compound, has been shown to impact a wide array of biological processes by interacting with multiple signaling pathways. Comprehensive in vitro and in vivo studies have highlighted PS’s effectiveness in inhibiting cell proliferation and promoting apoptotic mechanisms, and their detailed results are presented in Table 1 and illustrated in Figure 2.

The following sections provide a thorough overview of the anticancer activities attributed to PS, exploring its mechanisms of action and the specific signaling pathways it targets. These investigations have identified PS as a potent agent against a variety of cancer types, where it interrupts critical pathways involved in tumor growth and survival. Each cancer type responds differently, showcasing the adaptability and broad potential of PS as a key component in targeted cancer therapy.

### 7.1. PI3K/AKT Pathway

Phosphatidylinositol 3-kinase (PI3K/AKT/mTOR pathway cascade), or the PI3K/AKT pathway, is the most frequently altered pathway in humans for cancer development. This pathway plays a crucial role in several cellular processes, including cell cycle regulation, cell survival, metabolism, motility, angiogenesis, chemoresistance, and genomic stability [70]. The phosphorylation of Akt can trigger the subsequent phosphorylation of NF-kB, mTOR, and FoxO3a, which play a crucial role in cell growth, proliferation, survival, and angiogenesis [1]. Numerous research has yielded empirical data supporting the inhibitory efficacy of PS on the PI3K/AKT signaling pathway. In addition, PS treatment reduced the levels of mTOR-regulated phosphorylated S6K1 and phosphorylated 4E-BP1 [19,23,32,54].

Multiple biological processes are known to be linked to the PI3K/AKT pathway. Inhibiting AKT signaling reduces anti-apoptotic proteins while increasing pro-apoptotic molecules [71]. Previous studies have provided evidence indicating that PS can inhibit the transcription of FoxO3a-target genes, including cyclinD1 and Bcl-xL, while concurrently promoting the overexpression of the p21 and p27 genes [32]. PS also decreased angiogenesis by targeting many signaling pathways, including VEGF-induced AKT, ERK1/2, mTOR, and ribosomal protein S6 kinase [67]. These findings indicate that PS suppresses cell migration, invasion, and metastasis via affecting the PI3K/AKT pathway.

### 7.2. ROS Generation and MAPK Pathway

The three main subfamilies of mitogen-activated protein kinases (MAPKs) are extracellular signal-regulated kinases (ERKs), p38, and stress-activated protein kinases (JNK) [72]. According to published reports, PS increased phosphorylated JNK and p38 in K562 leukemic cells and BC while decreasing phosphorylated ERK [28,73]. Numerous research studies have shown that the activation of JNK is closely linked to the initiation of apoptosis in many cancers [25,28,33].

Reports were published that PS treatment leads to an increase in reactive oxygen species (ROS) generation, which in turn results in heightened mitochondrial permeability and a reduction in the mitochondrial membrane potential, which leads to cell death [21,23,24,28,71,74]. Moreover, it has been shown that increased levels of reactive oxygen species (ROS) have been associated with the buildup of unfolded proteins and endoplasmic reticulum (ER) stress, eventually resulting in cellular apoptosis [45]. ROS generated by PS triggers JNK activation. Numerous publications have shown that PS has the ability to trigger cellular apoptosis (Table 2) via the activation of the ROS/JNK signaling pathway [25,63,75,76]. According to recent research in colorectal cancer (CRC) cells, JNK inhibition [25,63,75,76] results in caspase suppression, which in turn causes apoptosis [45]. Furthermore, PS has been found to induce cell cycle arrest in the G1 phase, as well as apoptosis and autophagy [28]. These cellular responses were initiated by the ROS/ASK1/JNK signaling cascades.

### 7.3. NF-κB Pathway

The NF-kB transcription factors have been widely recognized for their role in the context of cancer [77]. The activation of NF-kB leads to the upregulation of NF-kB-dependent genes [78], and several other signaling pathways are thought to play important roles in this process. PS has been shown in multiple investigations to effectively suppress TNF-a or LPS-induced p-IKK, p-IkBα, and p65 translocation [35,44,46,63,69,79].

PS has significantly reduced the expression of NF-kB-dependent genes (those involved in apoptosis, invasion, and angiogenesis) and cytokines [19,20,32,35,44,46,63,69]. The inhibition of the AKT/mTOR pathway and NF-kB has also been shown to reduce tumor angiogenesis in osteosarcoma cells [80], which may improve the efficacy of anticancer medicines and radiation.

### 7.4. HIF-1a/SPHK-1 Pathway

Hypoxia, a common feature in advanced solid tumors, has been identified as a significant factor in the advancement of metastasis and the emergence of treatment resistance. The activation of hypoxia-inducible transcription factors facilitates the occurrence of this phenomenon [81,82]. The overexpression of the transcription factor known as hypoxia-inducible factor 1α (HIF-1α) in human cancers can be attributed to hypoxic conditions within the tumor. This event subsequently stimulates angiogenesis and augments the survival capabilities and proliferation of cancerous cells [83].

The inhibitory effects of PS have been seen in the context of hypoxia-induced sphere and colony growth. PS has been demonstrated to have an inhibitory effect on the activity of the cancer stem cell markers CD44, KLF4, OCT4, and AGO2. Similarly, other studies reported that PS had an inhibitory effect on many epithelial–mesenchymal transition (EMT) markers, including N-cadherin, fibronectin, vimentin, and ZEB1 [59,68].

SPHK1 stabilizes HIF-1a via a downstream pathway involving AKT and GSK-3b. The enzymatic activity of SPHK-1 increases in hypoxia, whereas ROS regulates its control. Several studies have shown that PS can potentially decrease the expression of HIF-1α via the inhibition of the SPHK-1 pathway. Further, sphingosine kinase-1 (SPHK-1) inhibition inhibited VEGF synthesis and cyclinD1 and CDK4 expression [1,68].

### 7.5. Ubiquitin–Proteosome Pathway

The ubiquitin–proteasome system coordinates physiological activities, including growth, death, cell cycle control, DNA repair, and antigen presentation [1]. The 26S eukaryotic proteasome comprises two 19S regulatory particles and one 20S catalytic particle. There are three catalytic sites on the 20S core particle: chymotrypsin-like (b5), trypsin-like (b2), and PGPH-like or caspase-like (b1) [84].

Different studies have shown that the conjugated ketone carbon (C6) of PS and the N-terminal threonine of the proteasomal b5 subunit have inhibitory effects on chymotrypsin-like activity. The inhibition results in polyubiquitination and subsequent modifications in the levels of Bax, p27, and IkBα, ultimately leading to the induction of cell death [39,43,85]. The suppression of Chk1 by PS has been found to cause a defect in DNA repair [86]. This could potentially broaden the use of olaparib in tumors that are proficient in BRCA but have TP53 mutations [86]. This suggests that PS can be used in combination with PARP inhibitor-based therapy.

Cancer cells exhibit increased expression of survivin, a protein that inhibits apoptosis and promotes cell survival by obstructing the mechanism of programmed cell death [1]. Several studies have shown that PS effectively decreases the levels of survivin by promoting its degradation via the ubiquitin–proteasome pathway, ultimately leading to apoptosis [37]. PS triggers apoptosis in prostate cancer cells via the generation of reactive oxygen species and the degradation of ubiquitin–proteasomes [24]. Additionally, PS decreases the invasiveness of breast cancer cells via enhancing RGS4 and suppressing proteasomal activity [41].

### 7.6. Wnt/β-Catenin Pathway

Wnt proteins function as critical mediators in a variety of important biological processes. The aberrant stimulation of the pathway mediated by Wnt/β-catenin can result in various illnesses, including cancers [1]. PS has shown the ability to modulate the Wnt/β-catenin pathway in MCF-7 breast cancer cells. This modulation is achieved by the targeting and downregulation of LRP6 expression and phosphorylation, resulting in a decrease in the total β-catenin levels. Additionally, PS treatment leads to an elevation in LC3-II levels, suggesting their potential involvement in the regulation of autophagy [54]. PS has also been seen to downregulate Wnt target genes, including cyclinD1, c-Myc, β-catenin, and cox-2, in HT-29 and HCT116 CRC cells [31].

Dishevelled (Dvl), a key component of Wnt/β-catenin signaling, increases β-catenin via phosphorylating/inactivating GSK3b and increasing AKT-Axin-GSK3b complex interaction [87]. Reports have shown that the degradation of β-catenin by GSK3b activation may cause PS-induced CRC cell death [31]. PS treatment decreased the p-AKT levels in MCF-7 breast cancer cells, suggesting that the PI3K/AKT pathway suppressed the Wnt/β-catenin pathway in breast cancer [54].

### 7.7. Shh/Gli1 Pathway

The sonic hedgehog (Shh)/glioma-associated oncogene homolog 1 (Gli1) signaling pathway is known to be aberrantly activated in various human neoplasms, playing a critical role in tumor development and progression. This pathway is divided into two branches: the canonical and the non-canonical pathways. In the canonical Shh pathway, tumor-produced Shh ligands negate the suppressive action of the Patched (PTCH) receptor on the Smoothened (SMO) protein. This de-repression of SMO initiates a cascade of molecular events that ultimately activates Gli1, a transcription factor responsible for modulating gene expression that affects numerous cellular processes critical to tumorigenesis [1]. A significant study by Lei et al. demonstrated that PS effectively disrupts this pathway. PS was shown to inhibit the nuclear localization of Gli1 in endothelial and pericyte cells within the tumor microenvironment. By preventing Gli1 from entering the nucleus, PS effectively shuts down the SHH/Gli1 signaling pathway, leading to a reduction in its ability to promote angiogenesis. This action results in the suppression of downstream signaling involved in the growth and maintenance of NCI-H1299 xenograft tumors, illustrating a potent anti-tumor mechanism of PS [34]. Through such targeted interference, PS holds promise as a therapeutic agent capable of mitigating the angiogenic and oncogenic potentials of the SHH/Gli1 pathway in cancer treatment.

### 7.8. Telomerase

Mammalian telomeres, which consist of a repetitive nucleotide sequence (TTAGGG), play a pivotal role in cellular aging and cancer biology. These sequences are capped and protected by a protein complex known as shelterin. This complex is critical for maintaining telomere integrity by preventing harmful end-to-end chromosomal fusions during the formation of the T-loop, a structure essential for telomere protection [1]. During cell division, these TTAGGG repeat sequences are progressively shortened, a process intrinsically linked to cellular aging known as senescence. This shortening mechanism serves as a natural limit to cell proliferation, thus acting as a barrier against the characteristic of uncontrolled cell growth in cancer. Notably, there is a strong correlation between the increased activity of the enzyme telomerase, which elongates telomeres, and the progression of cancer, underscoring the enzyme’s role in tumorigenesis [88]. Recent studies have highlighted the efficacy of PS in combating cancer by targeting telomere dynamics. PS has been shown to inhibit the expression and activity of human telomerase reverse transcriptase (hTERT), the catalytic subunit of telomerase, in both pancreatic ductal adenocarcinoma and prostate cancer cells. This inhibition is mediated through the downregulation of several key transcription factors, including Sp1, c-Myc, NF-kB, and STAT-3, which are crucial for hTERT transcriptional activation. Furthermore, PS impacts the PI3K/AKT pathway, a significant regulator of cell survival and proliferation. It inhibits the phosphorylation of AKT and subsequently hTERT, leading to a reduction in telomerase activity. This effect contributes to telomere shortening and promotes the induction of senescence, thus impeding the proliferation of cancer cells [36,38]. By disrupting these critical cellular pathways, PS not only impedes cancer progression but also has potential as a therapeutic agent in oncology.

## 8. Role of PS in Overcoming Multidrug Resistance and Enhancing Chemotherapeutic Efficacy

Multidrug resistance (MDR) represents a significant challenge in cancer therapy, as cancer cells can develop resistance to a variety of chemotherapeutic agents, each with distinct molecular structures and mechanisms of action. One of the key players in MDR is the ATP-binding cassette (ABC) transporter family, particularly ABCB1 (P-glycoprotein, Pgp), which functions as a potent drug efflux pump. The elevation of ABCB1, along with the multidrug resistance-associated protein 1 (MRP1/ABCC1) and the breast cancer resistance protein [89,90,91]. PS has shown potential in overcoming ABCB1-mediated chemotherapeutic drug resistance. In human oral epidermoid carcinoma cells (KBv200), PS has been found to disrupt ABCB1 stability independently of its mRNA expression, highlighting a novel approach to mitigating MDR [92]. Additionally, PS has demonstrated efficacy against imatinib-resistant chronic myeloid leukemia (CML) cells both in vitro and in vivo by inhibiting NF-κB and Bcr-Abl, key factors in the development of resistance [46]. Moreover, PS has been linked to the reversal of MDR in multidrug-resistant MCF-7/ADR breast cancer cells through its association with the AKT signaling pathway, emphasizing its role in broader MDR contexts [16].

The integration of natural products with conventional chemotherapeutics has emerged as a promising strategy to enhance anticancer efficacy [1]. The integration of natural products with conventional chemotherapeutics has emerged as a promising strategy to enhance anticancer efficacy. For instance, PS has shown synergistic effects with 5-fluorouracil (5-FU) in ESCC [63], and when combined with paclitaxel (taxol), it has enhanced anti-tumor activities through ROS-mediated mitochondrial dysfunction in cervical cancer cells [43]. Together, they inhibit ERK1/2 [15], triggering autophagy in human BC cells. Additionally, the combination of PS and cisplatin in A549 and NCI-H446 lung cancer cells has been effective in inducing apoptosis by inhibiting the AKT/GSK3β and miRNA-23a signaling pathways [47]. In pancreatic cancer, PS augments the cytotoxic effects of gemcitabine by blocking the NF-κB activation that is typically induced by gemcitabine treatment [14,35]. Furthermore, the co-administration of PS and bortezomib has shown cytotoxic effects against myeloma, illustrating the broad potential of PS in enhancing the effectiveness of multiple chemotherapeutic regimens [85].

## 9. Conclusions and Future Perspectives

The potential of Chinese herbal therapy, particularly PS, in cancer treatment is being increasingly recognized. PS’s effectiveness against cancer is mediated through several mechanisms, including the induction of autophagy, regulation of inflammation-related tumorigenesis, enhancement of chemosensitivity, and modulation of the tumor microenvironment and immune response. Its ability to inhibit cell proliferation, trigger apoptosis, prevent tumor metastasis and invasion, and suppress angiogenesis positions PS as a promising anticancer agent.

Despite the promising anticancer effects demonstrated by PS, significant research gaps remain that need to be addressed to advance PS from laboratory research to clinical applications. The extraction process of PS is inefficient, limiting its bioavailability and hindering further development. There is a critical need to understand the pharmacokinetics and pharmacodynamics of PS in diverse population groups. Detailed studies are required to explore how PS is metabolized and distributed in the human body, and how these processes vary by genetic makeup, age, and comorbid conditions. This understanding will be vital for optimizing dosages and minimizing potential side effects in clinical settings.

Furthermore, while PS has shown potential in modulating several cancer-related pathways, the exact molecular interactions and the potential for off-target effects remain underexplored. Future investigations should focus on the specificity of PS towards its molecular targets and its interaction with other cellular components in the tumor microenvironment. Such studies could elucidate the unintended impacts of PS, thereby refining its therapeutic profile. Additionally, comprehensive in vivo studies and clinical trials are needed to validate the efficacy and safety of PS observed in vitro. These studies should aim to establish robust clinical protocols and assess long-term outcomes to fully harness PS’s potential as a part of integrated cancer therapy. The integration of advanced technologies like CRISPR gene editing and AI-driven predictive models could further delineate the role of PS in cancer therapy, providing deeper insights and accelerating its clinical translation.

## Figures and Tables

**Figure 1 pharmaceuticals-17-00578-f001:**
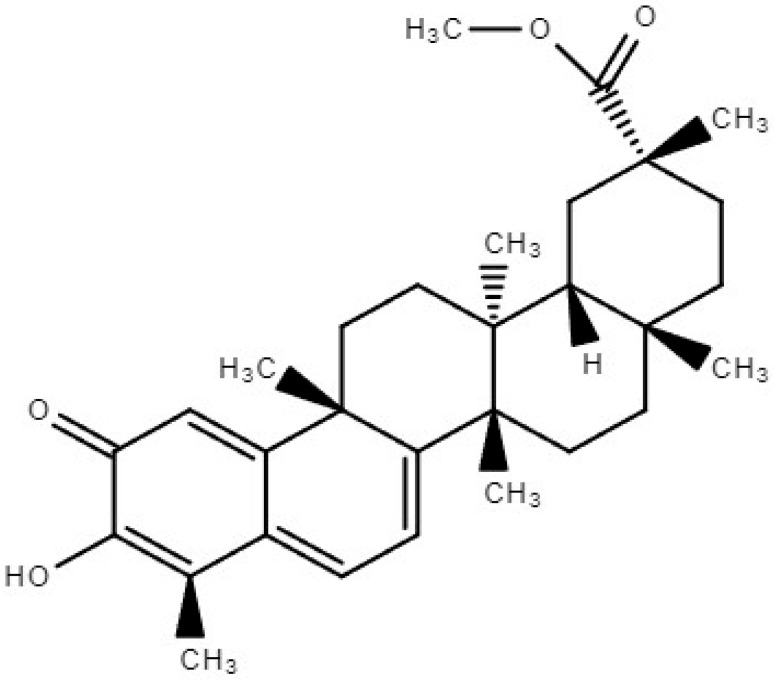
Chemical structure of Pristimerin.

**Figure 2 pharmaceuticals-17-00578-f002:**
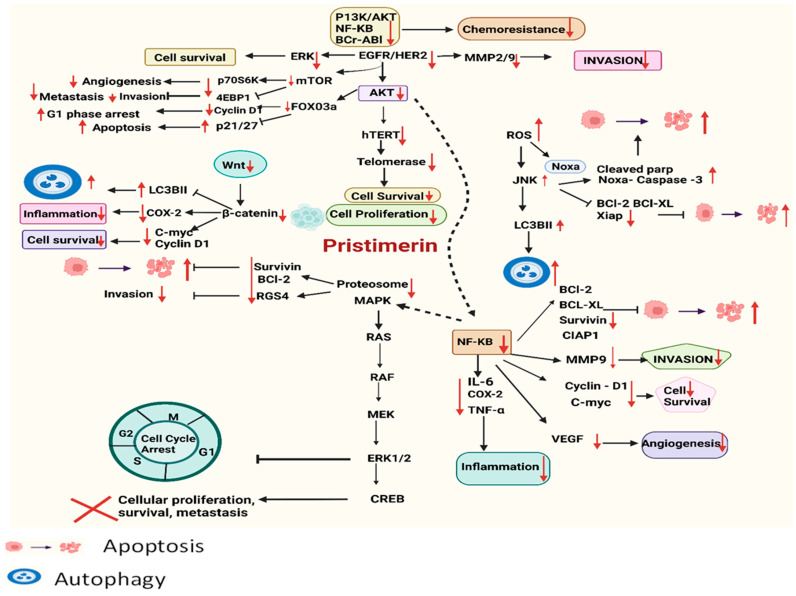
Schematic presentation of PS’s anticancer mechanisms of action targeting various signaling cascades via upregulation and downregulation of different regulatory proteins.

**Table 1 pharmaceuticals-17-00578-t001:** Anticancer activity of Pristimerin (PS) in in vitro models.

Cancer	Cell Lines	Mechanism of Action/s	PS Alone or/in Combination with Other Drug	References
Breast cancer	MDA-MB-231	Increased expression of RGS4 that suppressed migration and invasion	PS (0–3 µM)	[27]
	ADR-resistant MCF-7/ADR	Induced apoptosis via suppression of AKT pathway	PS (0–10 µM)	[16]
	MDA-MB-231	Enhanced autophagy induction	PS (5 µM) and paclitaxel (24 µM)	[15]
	MDA-MB-231, MDA-MB-468	Decreased levels of phosphorylated ASK1 and JNK	PS (0–0.6 µM)	[28]
	MCF-7, MDA-MB-231,4T1	Induced apoptosis, upregulated expression of miR-542-5p while downregulating DUB3 levels	PS (0–4 µM)	[29]
	epidermal growth factor receptor 2 (HER2)-positive SKBR3	Reduced fatty acid synthase and altered AKT, MAPK, and mTOR pathways	PS (0–10 µM)	[30]
Cervical cancer	HeLa	Activated ROS-dependent JNK, Bax, and PARP-1	PS	[25]
	HeLa, CasKi, SiHa	Induced ROS-dependent cell death	PS	[25]
Colorectal cancer	HCT116, HT-29	Inhibited Wnt/β-catenin signaling pathway	PS	[31]
	AOM/DSS model of colitis-associated colorectal cancer	Suppression of AKT and FOXO3 signaling	PS	[32]
Leukemia	HL-60	Induced apoptosis	PS	[26]
	K562	ROS-JNK-induced autophagy.	PS (0–0.8 µM)	[33]
Lung cancer	NCI-H1299	Targeted Shh/Gli1 signaling pathway	PS (0–500 nM)	[34]
	NCI-H1299	Decreased the rate of migration and invasion	PS (0–500 nM)	[34]
Ovarian cancer	OVCAR-5, MDAH- 2774, SK-OV-3	Inhibited AKT/NF-k B/mTOR pathway	PS (0–10 µM)	[19]
Pancreatic cancer	AsPC-1, BxPC-3, PANC-1	Abrogated Chk1/53BP1-mediated DNA repair, suppressed NF-κB activity	PS (200 nM) + gemcitabine (500 nM)	[14,35]
	MiaPaCa-2, Panc-1	Inhibited hTERT via suppressing transcription factors	PS (0–5 µM)	[36]
	MiaPaCa-2, Panc-1	Induced inhibition of AKT/NF-κB/mTOR pathway	PS (5 µM) + proteasome inhibitors	[20]
Prostate	LNCaP, PC-3	Decreased level of survivin and Bcl-2, and Inhibited hTERT mRNA expression levels.	PS (0–10 µM)	[24,37,38]
	PC-3, LNCaP, C4-2B	Induced apoptosis	PS (0–5 µM)	[39]
Oral squamous cell carcinoma	CAL-27, SCC-25	G1 arrest via inhibition of MAPK/Erk1/2 and AKT pathway	PS (0–1 µM)	[40]

**Table 2 pharmaceuticals-17-00578-t002:** Anticancer activity of Pristimerin (PS) in in vivo models.

Cancer	Animal Models	Mechanism of Action/s	PS Alone or/in Combination with Other Drug	References
Breast cancer	Breast cancer xenograft model	The inhibition of proteasomal activity and inhibition of tumor migration and invasion	PS (1 mg/kg)	[41]
	MDA-MB-231 tumor xenografts in nude mice	The stimulation of ROS/ASK1/JNK-mediated apoptosis and autophagy. Suppression of tumor growth	PS (0.5 mg/kg)	[28]
	Human breast cancer xenograft model	Suppression of VEGF, tumor growth, and angiogenesis	PS (3 mg/kg)	[42]
Cervical cancer	Tumor xenografts on nude mice	Apoptosis via mitochondrial signaling and activation of pro-apoptotic protein Bax	PS + Taxol	[43]
Colorectal cancer	Human colorectal cancer xenograft model	Inhibited tumor growth via targeting PI3K/AKT/mTOR pathway	PS (1 mg/kg)	[23]
	Human colorectal cancer xenograft model	Inhibited NF-кB signaling pathway	PS (1 mg/kg)	[44]
	Tumor xenograft in nude mice	Apoptosis through activation of the ROS/ER stress/JNK pathway	PS	[45]
Leukemia	Imatinib-resistant Bcr-Abl-T315I xenografts in mice	Suppressed TNFα-induced NFKB, as well as inhibition of the Bcr-Abl expression	PS (1 mg/kg)	[46]
Lung cancer	Human lung tumors xenograft model	Anticancer activity via targeting miR−23a/AKT/GSK3b pathway	PS (0.8 mg/kg) + cisplatin (2 mg/kg)	[47]
	Lung tissue samples from patients	Exerted anticancer activities through EphB4/CDC42/N-WASP pathway.	PS (0–8 µM)	[48]
Prostate cancer	Intra-tibial injection mouse model	Suppressed stem cell activity and angiogenesis via VEGF inhibition	PS (1.6 µM)	[49]

## Data Availability

Not applicable.

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
