# Peer review of "Anticancer Potential and Molecular Targets of Pristimerin in Human Malignancies"

_pharmaceuticals, 2024, doi:10.3390/ph17050578_

Round 1
Reviewer 1 Report
Comments and Suggestions for Authors
Perhaps authors could include botanical information about the origin of Pristmerin.
Authors could look into elaborating more in the text rather then referring to citation.
Authors should also consider reviewing the subject matter critically with originality rather then 'cut and merge' on some paragraphs.
Comments on the Quality of English LanguageLanguage usage is fairly good.
Author Response
Perhaps authors could include botanical information about the origin of Pristimerin.
AR: This information has been incorporated in the revised manuscript.
Authors could look into elaborating more in the text rather then referring to citation.
AR: We are very thankful to the reviewer for raising this issue. We have detailed the cited references in the revised manuscript.
Authors should also consider reviewing the subject matter critically with originality rather then 'cut and merge' on some paragraphs.
AR: We thank the reviewer for the valuable suggestion. The revised manuscript has been significantly improved, with most sections newly written and added information.
Reviewer 2 Report
Comments and Suggestions for Authors
Dear authors.
The problem of cancer is relevant for all countries of the world. There is no more effective therapy in the treatment of cancer than chemotherapy. But its side effect is significant, up to a fatal outcome. Research aimed at finding equally effective but less destructive approaches to cancer therapy, including with substances and components of alternative medicine, deserves attention. The topic touched upon in the article is relevant. The scientific content of the manuscript justifies its publication, but some additions and modifications will significantly improve the quality of the article.
Major comments:
1) In Abstract, the purpose of the study and the prospects for using the results obtained should be added.
Introduction
2) The purpose of the research and the scientific novelty should be clearly formulated.
3) Chinese medicine contains significant evidence of the proven effectiveness of various components, including in antitumor therapy. What determines the authors' interest in Pristimerin?
4) For each section, the authors' critical view of the existing problem of the use of Pristimerin in the treatment of cancer is missing. If such a significant number of confirmed positive effects of Pristimerin in cancer therapy have been established, why is it still not at the clinical trial stage?
5) Conclusions. The authors argue that more research is needed. What is supposed to be further investigated and why?
Author Response
The problem of cancer is relevant for all countries of the world. There is no more effective therapy in the treatment of cancer than chemotherapy. But its side effect is significant, up to a fatal outcome. Research aimed at finding equally effective but less destructive approaches to cancer therapy, including with substances and components of alternative medicine, deserves attention. The topic touched upon in the article is relevant. The scientific content of the manuscript justifies its publication, but some additions and modifications will significantly improve the quality of the article.
AR: We would like to express our sincere gratitude for your valuable comments and suggestions, which have guided us in improving our manuscript. We thank the reviewer for mentioning, "The topic touched upon in the article is relevant. The scientific content of the manuscript justifies its publication.” We have included the reviewer's suggestions in the revised manuscript. We have responded to the comments point by point as follows.
Major comments:
1) In Abstract, the purpose of the study and the prospects for using the results obtained should be added.
AR: The abstract has been rewritten as per the suggestions.
Introduction
2) The purpose of the research and the scientific novelty should be clearly formulated.
AR: The introduction section has been revised per the suggestions.
3) Chinese medicine contains significant evidence of the proven effectiveness of various components, including in antitumor therapy. What determines the authors' interest in Pristimerin?
AR: We sincerely appreciate your insightful comments and the opportunity to discuss the components of Chinese medicine used in antitumor therapy.
Our interest in pristimerin is specifically driven by its unique and promising properties as an anticancer agent. Studies have highlighted pristimerin’s efficacy in inducing apoptosis and inhibiting the proliferation of various cancer cell lines, which are not only fundamental mechanisms in cancer therapy but also areas where many treatments encounter resistance. Furthermore, pristimerin has shown potential in overcoming drug resistance, a critical hurdle in current cancer treatment paradigms, making it a compelling candidate for further investigation.
Moreover, pristimerin’s ability to synergize with existing chemotherapeutics enhances its value as a potential adjunct therapy, potentially improving efficacy and reducing side effects when used in combination. These distinctive features of pristimerin guided our choice to focus on this compound, believing it could contribute significantly to the advancement of cancer treatment strategies.
4) For each section, the authors' critical view of the existing problem of the use of Pristimerin in the treatment of cancer is missing. If such a significant number of confirmed positive effects of Pristimerin in cancer therapy have been established, why is it still not at the clinical trial stage?
AR: Thank you again for your constructive critique, which underscores the importance of elucidating the rationale behind specific research directions in the broader context of global health challenges.
We are grateful for your insightful critique, which points out the need to critically examine the progress and challenges in developing pristimerin as a cancer therapy. Your comment helped us clarify this crucial aspect in the revised manuscript.
Despite the promising preclinical data on pristimerin, several factors contribute to its current stage in the drug development pipeline. Firstly, the bioavailability and solubility of pristimerin pose significant challenges. As a naturally occurring quinonemethide triterpenoid, pristimerin exhibits poor water solubility, which complicates its formulation and delivery in biological systems—a common hurdle for many natural compounds transitioning into clinical phases.
Secondly, the pharmacokinetics of pristimerin, including its metabolism, distribution, and excretion, are not yet fully understood. These aspects are critical for ensuring that the compound reaches the target tissues in effective concentrations without eliciting toxicity. Extensive pharmacokinetic studies are required to determine the optimal delivery methods and dosing schedules.
Thirdly, the funding and interest for advancing natural compounds can often be limited, particularly when competing against synthetic molecules, which may offer more straightforward paths to patenting and commercialization. This economic aspect can slow the progression of natural compounds like pristine from the bench to the bedside.
5) Conclusions. The authors argue that more research is needed. What is supposed to be further investigated and why?
AR: Thank you for your thoughtful question regarding the need for further research on pristimerin as a cancer therapeutic. We recognize the importance of specifying the future research directions in our conclusions to underscore the rationale and critical needs in this area. Pristimerin has shown promising potential in preclinical studies, primarily due to its ability to target multiple pathways involved in cancer progression, such as cell cycle regulation, apoptosis, and metastasis. However, significant gaps remain that must be addressed through further research:
- Mechanism of Action: While current studies have identified several cellular targets and pathways affected by pristimerin, the exact mechanisms through which pristimerin exerts its anticancer effects are not fully understood. Detailed mechanistic studies are necessary to elucidate how pristimerin interacts with these molecular targets, which will aid in optimizing its efficacy and reducing potential side effects.
- Formulation and Delivery: The development of effective and safe delivery systems for pristimerin is critical due to its poor solubility and bioavailability. Research into nanoformulations or conjugation with polymers could provide viable methods to enhance its delivery to tumor sites, increasing its clinical applicability.
- Combination Therapies: Exploring the synergistic effects of pristimerin with other chemotherapeutic agents could provide insights into combination therapies that may enhance cancer treatment outcomes. Studies focusing on drug synergy could reveal lower dose requirements for pristine in, reducing toxicity while maintaining therapeutic efficacy.
- Toxicity and Safety Profiles: Comprehensive toxicological studies are essential to ensure the safety of pristimerin, especially for long-term use. Investigating its toxicity in different models, including various animal models and potentially in initial phase clinical trials, will be critical to advancing its development.
- Clinical Trials: Finally, initiating early-stage clinical trials is paramount to translate preclinical findings into clinical applications. These studies will provide critical data on dosing, pharmacokinetics, safety, and preliminary efficacy in humans.
In our revised manuscript, we expanded upon these points in the conclusion, highlighting the potential of pristimerin in cancer therapy and articulating a clear path for future research.
Reviewer 3 Report
Comments and Suggestions for Authors
This manuscript refers to a review on anticancer potential and molecular targets of Pristimerin, a quinonemethide triterpenoid which has been extracted from a variety of species of the Celastraceae and Hippocrateaceae families. Pristimerin's anticancer potential is widely reported and scientifically proven on different types of tumor cells through in vitro and in vivo studies. Some good reviews on the anticancer potential and molecular targets of Pristimerin have already been published previously, such as Chen et al. (2021), Li et al. (2019) and Yousef et al. (2017).
Li JJ, Yan YY, Sun HM, Liu Y, Su CY, Chen HB, Zhang JY. Anti-Cancer Effects of Pristimerin and the Mechanisms: A Critical Review. Front Pharmacol. 2019 Jul 12;10:746. doi: 10.3389/fphar.2019.00746. PMID: 31354475; PMCID: PMC6640652.
Yousef BA, Hassan HM, Zhang LY, Jiang ZZ. Anticancer Potential and Molecular Targets of Pristimerin: A Mini- Review. Curr Cancer Drug Targets. 2017;17(2):100-108. doi: 10.2174/1568009616666160112105824. PMID: 26758533.
Chen RZ, Yang F, Zhang M, Sun ZG, Zhang N. Cellular and Molecular Mechanisms of Pristimerin in Cancer Therapy: Recent Advances. Front Oncol. 2021 May 7;11:671548. doi: 10.3389/fonc.2021.671548. PMID: 34026649; PMCID: PMC8138054.
Although this topic is well studied and explored, a new review on anticancer potential and molecular targets of Pristimerin is welcome, as it will provide an opportunity to update on new discoveries and insights regarding this molecule.
This review was apparently well conducted; however, I highlight that, in general, the topics should be better explored. The authors draw on a few studies to discuss each topic and especially topics 4, 5 and 6. Please see the following articles:
Shu C, Yu X, Cheng S, Jing J, Hu C, Pang B. Pristimerin Suppresses Trophoblast Cell Epithelial-Mesenchymal Transition via miR-542-5p/EGFR Axis. Drug Des Devel Ther. 2020 Nov 2;14:4659-4670. doi: 10.2147/DDDT.S274595. PMID: 33173276; PMCID: PMC7646443.
Liu S, Dong Y, Wang Y, Hu P, Wang J, Wang RY. Pristimerin exerts antitumor activity against MDA-MB-231 triple-negative breast cancer cells by reversing of epithelial-mesenchymal transition via downregulation of integrin β3. Biomed J. 2021 Dec;44(6 Suppl 1):S84-S92. doi: 10.1016/j.bj.2020.07.004. Epub 2020 Jul 25. PMID: 35652598; PMCID: PMC9038948.
Cevatemre B, Erkısa M, Aztopal N, Karakas D, Alper P, Tsimplouli C, Sereti E, Dimas K, Armutak EII, Gurevin EG, Uvez A, Mori M, Berardozzi S, Ingallina C, D'Acquarica I, Botta B, Ozpolat B, Ulukaya E. A promising natural product, pristimerin, results in cytotoxicity against breast cancer stem cells in vitro and xenografts in vivo through apoptosis and an incomplete autopaghy in breast cancer. Pharmacol Res. 2018 Mar;129:500-514. doi: 10.1016/j.phrs.2017.11.027. Epub 2017 Dec 6. PMID: 29197639.
Zhang B, Zhang J, Pan J. Pristimerin effectively inhibits the malignant phenotypes of uveal melanoma cells by targeting NF‑κB pathway. Int J Oncol. 2017 Sep;51(3):887-898. doi: 10.3892/ijo.2017.4079. Epub 2017 Jul 25. PMID: 28766683.
Also, there are some points that need explanation and/or correction. Please see the comments below.
Initially, I highlight that the manuscript must be submitted to an extensive editing of English language. There are several spelling errors, unusual words from a technical-scientific point of view and sentences with extra spaces.
1. The abstract must be improved. The justification is not sustainable since Pristimerin is widely studied. The objectives must be revised since they are not robust enough to support the proposed review. Please see also:
- Line 21: “Quinone“ – please rephrase to: “quinone”;
- Line 22: “Portulacaceae” – wouldn't it be Hippocrateaceae?
- Line 24: please remove the underline;
- Line 25: “To be more specific, PS was discovered to have an effect...” – please rephrase to: “PS has an effect...”;
2. Keywords: Several keywords already appear in the title. Please replace these keywords by others to increase visibility of this review in databases.
3. Introduction:
- Line 35: “Cancer is a complex and formidable global health...” – formidable? – please review;
- Line 47: “for the formidable challenge of cancer...” - formidable? – please review;
- Line 53: “Pristimerin (PS), with its chemical formula C30H40O4...” – please provide the chemical name of Pristimerin in parentheses, followed by its molecular formula;
- Lines 53-54: “a quinone methide triterpenoid” – please rephrase to: “a quinonemethide triterpenoid”;
- Line 57: “P.grahami” – please rephrase to: “P. grahami”;
- Figure 1: Please improve the quality of the Figure 1;
- Line 78: “augment” – please rephrase to “increase”;
- Lines 79-81: please review the objectives based on the previous comment;
4. Other sections:
- Lines 95-96: “in vitro and in vivo...” – please provide the italic form and review the entire manuscript;
- Tables 1 and 2: please present the tables immediately after mentioning them in the text;
- Tables 1 and 2 must be improved and expanded considering that the anticancer activity of this molecule is widely studied;
- Line 179: “(PI3K)/AKT/mTOR pathway cascade)” – please rephrase to: “(PI3K/AKT/mTOR pathway cascade)”;
- Line 183: “The Akt phosphorylation pathway...” – please rephrase Akt to AKT and check the entire manuscript including the tables;
- Lines 183-186: this sentence is confusing – please review;
- Line 212: “... PS has the ability to trigger cellular apoptosis v (Table 2)via...” – please review - What is "v"?;
- Line 213: “...signaling pathway. [23, 51, 70, 71].” – please rephrase to: “...signaling pathway [23, 51, 70, 71].”;
- Line 213: please define “CRC” between parenthesis;
- Line 214: “...JNK[23, 51, 70, 71] inhibition results in ...” – please rephrase to: “...JNK inhibition [23, 51, 70, 71] results in ...”;
- Line 224: “... (those involved in apoptosis, invasion, angiogenesis)...” – please rephrase to: “...(those involved in apoptosis, invasion, and angiogenesis)...”;
- Line 263: “... In another study , it was shown that PS triggers apoptosis in ...” – please rephrase to: “... PS triggers apoptosis in ...”;
- Lines 265-266: “This sentence is confusing. I was unable to identify the relationship between Table 1 and reference [55] – please review;
- Lines 279 and 282: “...b-catenin” – please rephrase to ““...of β-catenin”;
- Line 280: “...In another study PS treatment decreased...” – please rephrase to “PS treatment decreased...”;
5. Conclusion: the conclusion presents a summary of the results. It should finalize the findings presented and point out perspectives for the advancement of knowledge in the area studied – please review.
6. References: References do not follow the format described in the “Instructions for authors”. Furthermore, the authors do not mention more recent works published in 2023 and 2024.
In my final comments, I recommend that the manuscript should be widely reviewed by the authors. The abstract, keywords, introduction and other sections must be rephrased to value the scientific advances for the area. I reinforce that the manuscript must be submitted to an extensive editing of English language.
Comments on the Quality of English LanguageExtensive editing of English language required.
Author Response
This manuscript refers to a review on anticancer potential and molecular targets of Pristimerin, a quinonemethide triterpenoid which has been extracted from a variety of species of the Celastraceae and Hippocrateaceae families. Pristimerin's anticancer potential is widely reported and scientifically proven on different types of tumor cells through in vitro and in vivo studies. Some good reviews on the anticancer potential and molecular targets of Pristimerin have already been published previously, such as Chen et al. (2021), Li et al. (2019) and Yousef et al. (2017).
Li JJ, Yan YY, Sun HM, Liu Y, Su CY, Chen HB, Zhang JY. Anti-Cancer Effects of Pristimerin and the Mechanisms: A Critical Review. Front Pharmacol. 2019 Jul 12;10:746. doi: 10.3389/fphar.2019.00746. PMID: 31354475; PMCID: PMC6640652.
Yousef BA, Hassan HM, Zhang LY, Jiang ZZ. Anticancer Potential and Molecular Targets of Pristimerin: A Mini- Review. Curr Cancer Drug Targets. 2017;17(2):100-108. doi: 10.2174/1568009616666160112105824. PMID: 26758533.
Chen RZ, Yang F, Zhang M, Sun ZG, Zhang N. Cellular and Molecular Mechanisms of Pristimerin in Cancer Therapy: Recent Advances. Front Oncol. 2021 May 7;11:671548. doi: 10.3389/fonc.2021.671548. PMID: 34026649; PMCID: PMC8138054.
Although this topic is well studied and explored, a new review on anticancer potential and molecular targets of Pristimerin is welcome, as it will provide an opportunity to update on new discoveries and insights regarding this molecule.
This review was apparently well conducted; however, I highlight that, in general, the topics should be better explored. The authors draw on a few studies to discuss each topic and especially topics 4, 5 and 6. Please see the following articles:
Shu C, Yu X, Cheng S, Jing J, Hu C, Pang B. Pristimerin Suppresses Trophoblast Cell Epithelial-Mesenchymal Transition via miR-542-5p/EGFR Axis. Drug Des Devel Ther. 2020 Nov 2;14:4659-4670. doi: 10.2147/DDDT.S274595. PMID: 33173276; PMCID: PMC7646443.
Liu S, Dong Y, Wang Y, Hu P, Wang J, Wang RY. Pristimerin exerts antitumor activity against MDA-MB-231 triple-negative breast cancer cells by reversing of epithelial-mesenchymal transition via downregulation of integrin β3. Biomed J. 2021 Dec;44(6 Suppl 1):S84-S92. doi: 10.1016/j.bj.2020.07.004. Epub 2020 Jul 25. PMID: 35652598; PMCID: PMC9038948.
Cevatemre B, Erkısa M, Aztopal N, Karakas D, Alper P, Tsimplouli C, Sereti E, Dimas K, Armutak EII, Gurevin EG, Uvez A, Mori M, Berardozzi S, Ingallina C, D'Acquarica I, Botta B, Ozpolat B, Ulukaya E. A promising natural product, pristimerin, results in cytotoxicity against breast cancer stem cells in vitro and xenografts in vivo through apoptosis and an incomplete autopaghy in breast cancer. Pharmacol Res. 2018 Mar;129:500-514. doi: 10.1016/j.phrs.2017.11.027. Epub 2017 Dec 6. PMID: 29197639.
Zhang B, Zhang J, Pan J. Pristimerin effectively inhibits the malignant phenotypes of uveal melanoma cells by targeting NF‑κB pathway. Int J Oncol. 2017 Sep;51(3):887-898. doi: 10.3892/ijo.2017.4079. Epub 2017 Jul 25. PMID: 28766683.
AR: We are very thankful to the reviewer for valuable suggestions and for providing some important references to be included in topics 4,5 and 6. Per the reviewer's recommendation, we have included these critical findings in the above-mentioned sections of the revised manuscript.
Also, there are some points that need explanation and/or correction. Please see the comments below.
Initially, I highlight that the manuscript must be submitted to an extensive editing of English language. There are several spelling errors, unusual words from a technical-scientific point of view and sentences with extra spaces.
AR: Thank you for your constructive feedback regarding the language quality of our manuscript. We acknowledge the critical importance of presenting our research clearly and accurately, and we appreciate your guidance in highlighting these concerns.
To address your comments, we have comprehensively revised the manuscript to correct the spelling errors, rectify the use of technically unusual words, and eliminate any extra spacing in the sentences. To ensure a high standard of academic writing, we took help from an expert who specializes in our field of study to review and refine our manuscript meticulously. This helped ensure that the terminology used was appropriate and consistent with current scientific norms.
Furthermore, we supplemented this review with the use of advanced language editing software, including a premium version of Grammarly, to provide an additional layer of editing. This combination of expert human review and technological assistance has greatly enhanced the readability and technical precision of our text.
We are confident that these efforts have significantly improved the manuscript, making it clearer and more accessible to readers and reviewers alike. We hope that these changes meet your expectations and the standards of the journal.
- The abstractmust be improved. The justification is not sustainable since Pristimerin is widely studied. The objectives must be revised since they are not robust enough to support the proposed review.
AR: Thank you for your comments. We have rewritten the abstract to include your suggestions.
Please see also:
- Line 21: “Quinone“ – please rephrase to: “quinone”;
AR: Thank you for your comment. This has been rephrased in the revised manuscript.
- Line 22: “Portulacaceae” – wouldn't it be Hippocrateaceae?
AR: We apologize for the typographical mistake. It has now been corrected in the revised manuscript.
- Line 24: please remove the underline;
AR: The Underline has been removed.
- Line 25: “To be more specific, PS was discovered to have an effect...” – please rephrase to: “PS has an effect...”;
AR: Thank you for your valuable comment; necessary corrections have been made in the revised version.
- Keywords: Several keywords already appear in the title. Please replace these keywords by others to increase visibility of this review in databases.
AR: Thank you for your suggestion. We have replaced the keywords to increase the visibility of our review in the database.
- Introduction:
- Line 35: “Cancer is a complex and formidable global health...” – formidable? – please review;
AR: The sentence in the revised manuscript has been corrected.
- Line 47: “for the formidable challenge of cancer...” - formidable? – please review;
AR: The sentence in the revised manuscript has been corrected.
- Line 53: “Pristimerin (PS), with its chemical formula C30H40O4...” – please provide the chemical name of Pristimerin in parentheses, followed by its molecular formula;
AR: The revised manuscript has added the chemical name (20α-3-hydroxy-2-oxo-24-nor-friedela-1-10,3,5,7-tetraen-carboxylic acid-29-methyl ester).
- Lines 53-54: “a quinone methide triterpenoid” – please rephrase to: “a quinonemethide triterpenoid”;
AR: The changes have been included in the revised manuscript.
- Line 57: “P.grahami” – please rephrase to: “P. grahami”;
AR: The changes have been included in the revised manuscript.
- Figure 1: Please improve the quality of the Figure 1;
AR: Thank you for your comment. The quality of Figure 1 has now been improved in the revised manuscript.
- Line 78: “augment” – please rephrase to “increase”;
AR: The changes have been included in the revised manuscript.
- Lines 79-81: please review the objectives based on the previous comment;
AR: The objectives have been reviewed and edited in the revised manuscript.
- Other sections:
- Lines 95-96: “in vitro and in vivo...” – please provide the italic form and review the entire manuscript;
AR: Thank you for your valuable comment. The changes have been included in the revised manuscript.
- Tables 1 and 2: please present the tables immediately after mentioning them in the text;
AR: Thank you for your comment. Tables 1 and 2 are presented after mentioning them in the text.
- Tables 1 and 2 must be improved and expanded considering that the anticancer activity of this molecule is widely studied;
AR: Thank you for your valuable comment. The dosage of PS/combination has been included in the revised tables.
- Line 179: “(PI3K)/AKT/mTOR pathway cascade)” – please rephrase to: “(PI3K/AKT/mTOR pathway cascade)”;
AR: In the revised manuscript, “(PI3K)/AKT/mTOR pathway cascade)” has now been rephrased to (PI3K/AKT/mTOR pathway cascade).
- Line 183: “The Akt phosphorylation pathway...” – please rephrase Akt to AKT and check the entire manuscript including the tables;
AR: The changes have been included in the revised manuscript.
- Lines 183-186: this sentence is confusing – please review;
AR: The sentence has been rephrased in the revised manuscript.
- Line 212: “... PS has the ability to trigger cellular apoptosis v (Table 2)via...” – please review - What is "v"?;
AR: We apologize for the mistake. This has been corrected in the revised manuscript.
- Line 213: “...signaling pathway. [23, 51, 70, 71].” – please rephrase to: “...signaling pathway [23, 51, 70, 71].”;
AR: The changes have been included in the revised manuscript.
- Line 213: please define “CRC” between parenthesis;
AR: The “CRC” has been defined between the parenthesis in the revised manuscript.
- Line 214: “...JNK[23, 51, 70, 71] inhibition results in ...” – please rephrase to: “...JNK inhibition [23, 51, 70, 71] results in ...”;
AR: The changes have been included in the revised manuscript.
- Line 224: “... (those involved in apoptosis, invasion, angiogenesis)...” – please rephrase to: “...(those involved in apoptosis, invasion, and angiogenesis)...”;
AR: The changes have been included in the revised manuscript.
- Line 263: “... In another study , it was shown that PS triggers apoptosis in ...” – please rephrase to: “... PS triggers apoptosis in ...”;
AR: Corrections have been included in the revised manuscript.
- Lines 265-266: “This sentence is confusing. I was unable to identify the relationship between Table 1 and reference [55] – please review;
AR: We thank you for pointing out this. We have reviewed and edited the sentence accordingly.
- Lines 279 and 282: “...b-catenin” – please rephrase to ““...of β-catenin”;
AR: “...b-catenin” has been rephrased to ““...of β-catenin’’ in the revised manuscript.
- Line 280: “...In another study PS treatment decreased...” – please rephrase to “PS treatment decreased...”;
AR: The changes have been included in the revised manuscript.
- Conclusion: the conclusion presents a summary of the results. It should finalize the findings presented and point out perspectives for the advancement of knowledge in the area studied – please review.
AR: Thank you for your valuable feedback. We have revised our conclusion.
- References:References do not follow the format described in the “Instructions for authors”. Furthermore, the authors do not mention more recent works published in 2023 and 2024.
AR: The references have been revised according to the journal's instructions. Limited studies published in 2023 and 2024 have been incorporated into the revised manuscript.
In my final comments, I recommend that the manuscript should be widely reviewed by the authors. The abstract, keywords, introduction and other sections must be rephrased to value the scientific advances for the area. I reinforce that the manuscript must be submitted to an extensive editing of English language.Comments on the Quality of English Language: Extensive editing of English language required.
AR: We appreciate your insightful feedback and recommendations. To address your comments, we have comprehensively revised the manuscript to correct the spelling errors, rectify the use of technically unusual words, and eliminate any extra spacing in the sentences. To ensure a high standard of academic writing, we took help from an expert who specializes in our field of study to review and refine our manuscript meticulously. This helped ensure that the terminology used was appropriate and consistent with current scientific norms.
Furthermore, we supplemented this review with the use of advanced language editing software, including a premium version of Grammarly, to provide an additional layer of editing. This combination of expert human review and technological assistance has greatly enhanced the readability and technical precision of our text.
Reviewer 4 Report
Comments and Suggestions for Authors
After reading the article Anticancer Potential and Molecular Targets of Pristimerin in Human Malignancies, I found the information presented by the authors very interesting. I only have a few minor suggestions.
1. Please attach in the tables the doses that were used of PS.
2. Please check the references, since the name of the journal must be in italics, the year in bold and put the doi in the references that have it.
Comments on the Quality of English LanguageMinor editing of English language required
Author Response
After reading the article Anticancer Potential and Molecular Targets of Pristimerin in Human Malignancies, I found the information presented by the authors very interesting. I only have a few minor suggestions.
AR: We appreciate your positive feedback.
- Please attach in the tables the doses that were used of PS.
AR: Thank you for the suggestions. We have revised and incorporated the PS doses in Tables 1 and 2 of the revised manuscript.
- Please check the references, since the name of the journal must be in italics, the year in bold and put the doi in the references that have it. Comments on the Quality of English Language: Minor editing of English language required
AR: We appreciate your insightful feedback and recommendations. The references have been revised according to the standards provided by the journal, and the paper has undergone thorough editing to ensure grammatical accuracy and proper use of the English language.
Round 2
Reviewer 2 Report
Comments and Suggestions for Authors
Dear Authors
My comments have been taken into account. But the format of the list of references and the quality of the drawings require attention.
Reviewer 3 Report
Comments and Suggestions for Authors
All comments and suggestions were accepted and implemented.